# Unveiling coherently driven hyperpolarization dynamics in signal amplification by reversible exchange

Jacob R. Lindale [1], Shannon L. Eriksson[1], Christian P.N. Tanner[1], Zijian Zhou[1], Johannes F.P. Colell[1], Guannan Zhang[1], Junu Bae [1], Eduard Y. Chekmenev[2,3], Thomas Theis [1] & Warren S. Warren[4]

Signal amplification by reversible exchange (SABRE) is an efficient method to hyperpolarize spin-1/2 nuclei and affords signals that are orders of magnitude larger than those obtained by thermal spin polarization. Direct polarization transfer to heteronuclei such as $^{13}C$ or $^{15}N$ has been optimized at static microTesla fields or using coherence transfer at high field, and relies on steady state exchange with the polarization transfer catalyst dictated by chemical kinetics. Here we demonstrate that pulsing the excitation field induces complex coherent polarization transfer dynamics, but in fact pulsing with a roughly 1% duty cycle on resonance produces more magnetization than constantly being on resonance. We develop a Monte Carlo simulation approach to unravel the coherent polarization dynamics, show that existing SABRE approaches are quite inefficient in use of para-hydrogen order, and present improved sequences for efficient hyperpolarization.

[1] Department of Chemistry, Duke University, Durham, NC 27708, USA. [2] Department of Chemistry, Karmanos Cancer Institute (KCI), Wayne State University, Detroit, MI 48202, USA. [3] Russian Academy of Sciences, Leninskiy Prospekt 14, Moscow, Russia 119991. [4] Department of Physics, Chemistry, Biomedical Engineering, and Radiology, Duke University, Durham, NC 27708, USA. These authors contributed equally: Jacob R. Lindale, Shannon L. Eriksson. Correspondence and requests for materials should be addressed to W.S.W. (email: warren.warren@duke.edu)

While nuclear magnetic resonance (NMR) experiments boast a wealth of information, low thermal spin magnetization significantly limits sensitivity. Large magnetic fields increase the Zeeman interaction energy, thus inducing higher polarizations, but realistic field strengths still imply very small fractional magnetization near room temperature. Such issues may be circumvented by hyperpolarization methods, which afford significantly higher nuclear polarization and provide signals that are orders of magnitude larger than those achieved with thermal polarization. DNP[1–6], CIDNP[7–9], PHIP[10–17], and SABRE[18–32] are examples of such hyperpolarization modalities. Indeed, hyperpolarization techniques[33] are now having a significant impact on biomedical imaging[34–36].

Signal Amplification By Reversible Exchange (SABRE), which utilizes an iridium polarization transfer catalyst (PTC), is a convenient non-hydrogenative PHIP variant because it is inexpensive, only requires simple experimental hardware, and can be repeated many times without consuming substrate. In fact, unlike other hyperpolarization techniques, SABRE can be employed for continuous hyperpolarization of imaging agents[37]. Importantly, SABRE provides high levels of polarization (i.e., $P > 15\%$) in under a minute[31,32,38]. During the SABRE process, the PTC establishes scalar couplings between $p$-$H_2$ (which acts as a polarization source) and the target ligand. Under the right circumstances, these couplings convert hydrogen singlet order to flow into magnetization on the target ligands, including the very interesting case of direct transfer to heteronuclei such as $^{15}N$. This flow of order is achieved by level anti-crossings at low (micro-Tesla) field (SABRE-SHEATH and later variants) or at high field in rotating frames created by very weak rf irradiation (LIGHT-SABRE and later variants)[19,21,24,25,30,39–42]. In either case, the system evolves under the given spin Hamiltonian until the PTC dissociates (either losing a ligand or losing $p$-$H_2$) and rebinding of unpolarized ligands from solution can create additional magnetization if the order in the hydrogen has not been depleted.

In typical heteronuclear experiments, the coupling between the bound parahydrogen atoms (which tends to preserve the singlet order), the couplings between those atoms and iridium-bound nitrogen in the PTC complex (which lets the order flow into nitrogen magnetization), and the width of the exchange-broadened resonance of the active complex are all very similar (10–30 Hz). This makes the SABRE process fundamentally incoherent, and impossible to model correctly by traditional approaches such as inserting relaxation into the density matrix. However, we demonstrate here that pulsing the field that generates hyperpolarization restores observable coherent hyperpolarization dynamics in both the low-field and high-field limits, providing a route to significantly boost hyperpolarized signals and efficiently consume singlet order. Furthermore, we develop and implement a Monte Carlo (MC) simulation approach to understand the coherent hyperpolarization dynamics. The MC simulations require fewer assumptions about the system and outperform current models[19,20] in the prediction of experimental data.

## Results

**Spin dynamics of SABRE.** In the SABRE-SHEATH experiment, the spin Hamiltonian is given as the sum of the Zeeman interaction and the scalar coupling terms in the strong coupling limit like:

$$\hat{H}_{\text{low-field}} = \omega_{iI} \sum_{i=1}^{n} \hat{I}_{z,i} + \omega_{iS} \sum_{i=1}^{n} \hat{S}_{z,i} + 2\pi \sum_{i \neq j} \mathcal{J}_{ij} \hat{I}_i \cdot \hat{I}_j + 2\pi \sum_{i \neq j} \mathcal{J}_{ij} \hat{S}_i \cdot \hat{S}_j + 2\pi \sum_{i \neq j} \mathcal{J}_{ij} \hat{I}_i \cdot \hat{S}_j \tag{1}$$

In this notation, $\omega_{iI}$ is the Larmor frequency of the hydrogen atoms ($\approx 42$ Hz at 1 $\mu$T), $\omega_{iS}$ is the Larmor frequency of the heteroatoms ($\approx 4$ Hz at 1 $\mu$T for $^{15}N$), and $J_{ij}$ is the scalar coupling between the spins $i$ and $j$. In high-field hyperpolarization experiments (where here for simplicity we illustrate only irradiation at the heteronuclear frequency), the secular approximation is taken with the heteronuclear scalar couplings, truncating the spin product $\hat{I}_i \cdot \hat{S}_j$ to the $z$-terms, and an $x$-phase pulse is introduced in the form of

$$\hat{H}_{\text{high-field}} = \sum_{i=1}^{n} \Omega_i \hat{S}_{z,i} + 2\pi \sum_{i \neq j} \mathcal{J}_{ij} \hat{I}_i \cdot \hat{I}_j + 2\pi \sum_{i \neq j} \mathcal{J}_{ij} \hat{I}_{i,z} \hat{S}_{j,z} + \omega_1 \sum_{i=1}^{n} \hat{S}_{x,i} \tag{2}$$

where $\omega_1$ is the nutation frequency of the pulse and $\Omega$ is the offset from resonance.

Once magnetic contact is established between the nuclei of the ligand and the parahydrogen singlet order via the PTC, the spin density $\hat{\rho}$ evolves coherently under the Liouville von Neumann (LvN) equation

$$\partial_t \hat{\rho}(t) = -i[\hat{H}, \hat{\rho}(t)] \tag{3}$$

until the complex dissociates. Given the random distribution of PTC dissociation events, it would be easy to motivate dynamics that appear as coherent Rabi oscillations becoming pumped to saturation and converging to an average, reduced spin density[19]. While in certain limits, we have found this limit to be a valid description of the evolution of the SABRE dynamics, it is not the situation for a vast majority of the systems. Firstly, because $J_{NH}$ is significantly large with respect to typical PTC lifetimes (20–50 ms), coherent evolution does not provide small perturbations to the overall dynamics, as would be required for such a perturbative, averaging method. In fact, all methods that utilize ensemble-averaged equations of motion will fail to correctly describe the dynamics in this exchange regime, as discretized dissociation events and subsequent evolution of the system will generate larger excursions from the average density matrix than would be allowed by conventional methods. For the same reasons, the superoperator models for SABRE, such as that by Knecht et al.[20], will also fail to correctly describe the hyperpolarization dynamics. However, by simply numerically simulating discrete dissociation events of individual PTCs in a Markov Chain Monte Carlo fashion and performing ensemble averaging of the resultant spin dynamics solution (see Supplementary Notes 1 and 4 for details), instead of the equation of motion, we arrive at a mathematically and physically fair model of the coherent hyperpolarization dynamics.

**Coherent hyperpolarization dynamics.** Using the form of the spin Hamiltonian presented in Eqs. 1 and 2, along with the computational method described above, we have constructed two SABRE experiments (Fig. 1) by which the coherent hyperpolarization dynamics may be studied and used to provide significant signal boosts over previously reported techniques. These are the (high field) Delayed Adiabatic Ramps Transfer Hyperpolarization (Fig. 1a), or DARTH-SABRE, and (low field) coherent SHEATH (Fig. 1b) experiments. Like other Spin Lock Induced Crossing (SLIC)-based methods for hyperpolarization[21,25], the DARTH-SABRE experiment only works for systems where the hydrides are chemically but not magnetically equivalent; we have chosen to test the DARTH-SABRE experiment using the canonical 4-spin AA'XX' $[Ir(H)_2(IMes)(^{15}N\text{-pyr})_3]^+$ SABRE system (Fig. 1a). The coherent SHEATH experiment does not have the same spin-topology restrictions that the DARTH-SABRE experiment does, and as such, we have chosen to polarize the 3-spin AA'X [Ir

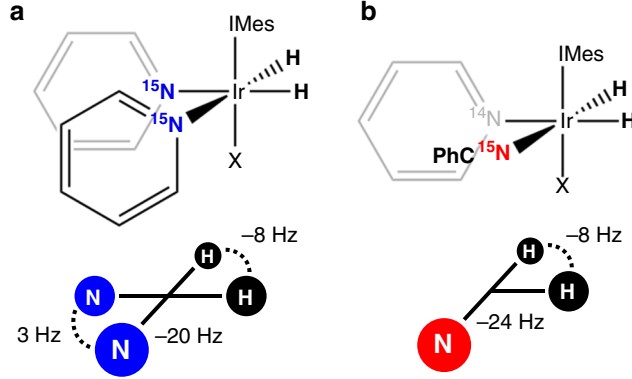

**Fig. 1** Model SABRE systems. **a** DARTH-SABRE (4-spin AA'XX' system) and **b** coherent SHEATH (3-spin AA'X system) experiment. The magnetic structure of each system is shown below the chemical structure along with its homonuclear (dashed) and heteronuclear (solid) J-couplings

$(H)_2(IMes)(^{15}N\text{-}PhCN)(pyr)_2]^+$ (PhCN = benzonitrile) SABRE system (Fig. 1b).

In the DARTH-SABRE experiment, pulses that are slightly off-resonance from the bound $^{15}N$-pyridine spin are applied while ramping the pulse power from an initial value of $\omega_1 = 32$ Hz with a rate of 42 Hz/s, directly hyperpolarizing z-magnetization (see Supplementary Note 3). This ramp induces hyperpolarized signals that are ~20% larger than if the pulse was a simple square pulse. For a fully enriched system like this, a $90_x$ $^1$H-pulse (Fig. 2a) is used to refocus hydride singlet order into a state that would not destroy the necessary initial $\beta$-magnetization from thermal ligands if polarizing the $T_N^{+1}$ state and vice versa, which gave an average of 2× larger signals than without the refocusing pulse (see Supplementary Note 3). Furthermore, the experimental data match the predicted DARTH dynamics with excellent agreement (Fig. 2c) with an AA'XX' spin system, exhibiting a strongly rising exchange baseline (the polarization to which the dynamics converge) to the dynamics and quite significantly damped coherent oscillations, due to the relatively short lifetime of the $^{15}N$-pyridine complex (20 ms). While the enhancements shown here are modest, they become exponentially greater with shorter delay times, producing a maximum enhancement of $\varepsilon = 1350$ (Fig. 2e).

Similar to the DARTH-SABRE experiment, the coherent SHEATH experiment pulses a microTesla evolution field of ~0.6 μT to allow hyperpolarization transfer while interleaving storage fields that are approximately 100-fold greater than the evolution field (see Supplementary Note 2). This ensures no coherent evolution of the spin system during the delay. For the AA'X spin system used in these experiments, the data match the predicted SHEATH dynamics with exceptional accuracy (Fig. 2d) and result in an approximately 2.5-fold enhancement of signals that are coherently hyperpolarized over the exchange baseline, which is synonymous with the steady-state SABRE signal, and produces $^{15}N$ polarizations of ~4.5%. The data indicate that the natural-abundance pyridine co-ligand has no observable effect on the coherent component of the SABRE dynamics, most likely due to minimal coupling into the system by the small $^4J_{HH}$ couplings from the para-hydrogen derived hydrides to the ortho-proton on the pyridine. This is very important, as it means that the effect of binding a natural-abundance co-ligand, like pyridine in this complex, only affects chemical dynamics and thus is incorporated by simply changing the exchange rate. The $^{14}N$ nucleus in these experiments will not affect the dynamics of these experiments, as the quadrupolar relaxation of the $C_1$-symmetric complex is very fast (sub-millisecond).

The most significant difference in the DARTH and SHEATH dynamics is the exchange baseline, which is readily explained by the evolution of the singlet population (Fig. 2f) under these conditions and arises from the continual rebinding of unpolarized ligands after the previous species dissociates. The DARTH dynamics exhibit a significantly stronger exchange baseline as the initial consumption of the singlet order is consistently less by nearly a factor of 2 than the SHEATH dynamics, meaning that considerably more polarization may be generated by successively associating ligands. This effect arises as it is the secular term $(\hat{I}_{i,z}\hat{S}_{j,z})$ which drives the hyperpolarization dynamics, depreciating the magnitude of the state couplings with respect to a SHEATH Hamiltonian, where dynamics are driven by the non-secular $\hat{I}_i^{\pm}\hat{S}_j^{\mp}$ terms. Under the correct experimental conditions, significant population transfer may still be induced at high field while minimizing the singlet order consumption. The systems shown here exemplify two limits to SABRE dynamics, and, for instance, indicate that all SHEATH experiments should be performed with pulsed excitation but that hyperpolarization efficiency is generally greater for DARTH experiments under the correct conditions.

The systems and simulations shown in Fig. 2 all assume that the sample composition is either fully $^{15}N$ labeled (forming four spin-1/2 AA'XX' systems) or that the labeled ligand concentration is significantly greater than the unlabeled ligand concentration (forming the spin-1/2 AA'X systems). However, the simulations are readily extensible to the case where the target ligand is isotopically dilute and is exemplified by DARTH dynamics of such samples. In this case, we assume that most of the PTCs at time $t = 0$ do not contain polarizable ($^{15}N$) ligands, as $^{14}N$-pyridine is significantly off-resonance from the DARTH pulse and will not generate hyperpolarization. However, the $^{14}N$-pyridine must be included in the spin system as auxiliary nucleus, as the quadrupolar relaxation time $T_Q$ in a $C_{2v}$-symmetric complex, like the tris-pyridine complex used in the DARTH experiments, is known to be longer[43] than free pyridine. The $^{14}N$ resonance of the $[Ir(H)_2(IMes)(^{14}N\text{-}pyr)_3]^+$ complex is considerably broader than that of the $^{15}N$ resonance, with $T_Q = 2.2$ ms, and provides a relaxation mechanism that makes the singlet order of the hydrides significantly less effective at long times. This is the simplest chemical example of a co-ligand which alters the coherent-component of the dynamics, highlighting the extensibility of the QMC simulations to study a rich variety of SABRE systems. When $^{15}N$-ligands do bind, the dynamics are dominated by species with one bound, labeled ligand (forming four spin AA'XQ systems, where Q = $^{14}N$-pyridine); and that when a polarized ligand dissociates, it is usually replaced with an unpolarizable ligand, hence the $^{15}N$-magnetization does not evolve under a DARTH pulse. Depending on sample composition and dissociation rates of the various ligands, the average time where the catalyst is hyperpolarization inactive, here called $T_{inactive}$, is often important for the dynamics.

If $T_{inactive}$ is long with respect to the hydride exchange and the inter-pulse delay is short, the DARTH-SABRE experiment becomes quasi-CW, for which it is sufficient to simulate the system as if the PTCs only bind a hyperpolarization active ligand once (Fig. 3) as each active PTC will, on average, only see a single pulse and the probability of initializing at a time other than $t = 0$ is negligible. This effect changes when $T_{inactive}$ is shorter than the hydride exchange, in which case initialization must be allowed at any time during the dynamics (Fig. 3). The length of $T_{inactive}$ may be motivated from statistics and is taken to be $T_{inactive} = (k_{d,pyr}\zeta)^{-1}$, where $\zeta$ is the $^{15}N$-enrichment.

When comparing the isotopically enriched samples to the natural abundance samples, the inclusion of a $T_{inactive} = 12$ s accounts for the deviation of the re-binding dynamics from the

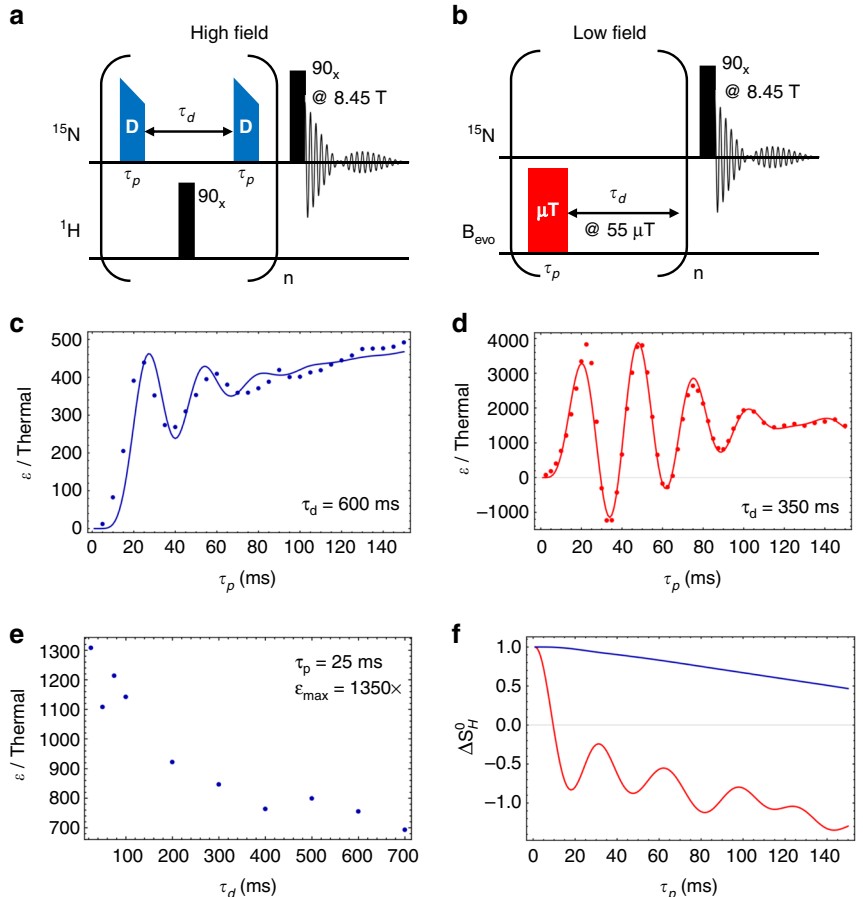

**Fig. 2** Coherent hyperpolarization experiments. **a** DARTH_SABRE pulse sequence, where DARTH-pulses of length $\tau_p$ are given at intervals of $\tau_d$ for "n" repetitions and applied to the bound $^{15}$N spin slightly off-resonance. The pulses adiabatically ramp towards the optimal matching condition to induce higher polarizations. **b** Coherent SHEATH pulse sequence, where analogous pulses are delivered to the sample at the optimal field condition, which is typically ≈0.5 µT, with inter-pulse delays using a ≈55 µT field to store magnetization. **c** Experimental DARTH-SABRE dynamics of an AA'XX' spin system using 50 mM $^{15}$N-pyridine with an inter-pulse delay of $\tau_d$=600 µs at 8.45 T. **d** Experimental coherent SHEATH dynamics of an AA'X spin system using 100 mM $^{15}$N-benzonitrile with 33 mM pyridine and an inter-pulse delay of $\tau_d$ = 350 ms; detection was performed at 8.45 T. Data are shown fit to numerical QMC simulations using average PTC lifetimes of 20 ms (DARTH) and 50 ms (SHEATH). **e** Polarization as a function of the delay parameter for the DARTH-SABRE sequence using a 25 ms DARTH pulse, which corresponds to a π pulse. **f** Evolution of singlet population excess (DARTH AA'XX': $S_H^0$–$T_H^0$, blue line; SHEATH AA'X: $S_H^0$–$T_H^-$, red line) under the conditions used in the fit of the experimental data and normalized to the initial $S_H^0$ population

isotopically enriched sample (a lack of a significant exchange baseline), suggesting that sample composition is a critical factor to consider when deciding to run a pulsed experiment like DARTH-SABRE or a static-field experiment like SABRE-SHEATH. The highest overall enhancements are achieved in this fractionally labeled regime, and the highest relative polarizations are observed in the coherent dynamics regime (~20 ms DARTH-pulse). The re-binding effect may be recovered, as predicted from the simulations, by lengthening the delay time to 600 ms (Fig. 3). A 14% $^{15}$N-enriched sample was prepared to mimic the binding dynamics of the natural abundance system but with greater signal-to-noise. For this system, $T_{inactive} = 350$ ms is sufficiently long for the experimentally observed window to reproduce the same effect while maintaining the condition that $T_{inactive} < k_{d,H2}^{-1}$. To confirm that this sample adequately reproduced the binding dynamics of the natural abundance system, the dynamics were monitored both with and without the $^1$H-refocusing pulse, which confirmed that $T_{inactive}$ of the order of the lifetime of the hydrides as there was no additional enhancement observed.

SABRE is also very important for the hyperpolarization of both $^1$H and $^{13}$C, by which the mechanism is either direct polarization

via $J$-couplings between the target nucleus and the $p$-H$_2$ derived hydrides or by spin-relay from a $^{15}$N nucleus. As evolution in the QMC simulation is performed fully quantum mechanically, it is trivial to examine the hyperpolarization dynamics of these other nuclei and arbitrarily complex systems up to 15-spins may be studied. This size restriction only arises as this is the limit of the size of the Hamiltonian that can be exponentiated exactly in Hilbert space, which is how the QMC simulations are constructed. In either case, the construction of the simulation is (in effect) no different than as described above, with only alterations in the quantum part of the spin Hamiltonian to incorporate these nuclei and ensuring that all nuclei associated with one ligand dissociate simultaneously. These systems are often of greater complexity than the simple 3-spin and 4-spin systems shown previously. In fact, the simulation of the ortho-$^1$H line-shape on the $^{15}$N-pyridine ring after a single 25 ms DARTH pulse requires the construction of an 8-spin AA'(XB$_2$)(X'B'$_2$) system (Fig. 4). Instead of recording the polarization at each step, the density matrix of each dissociating pyridine was ensemble averaged using the QMC routine, which was then used to seed a simple pulse-acquire routine to generate the $^1$H spectrum

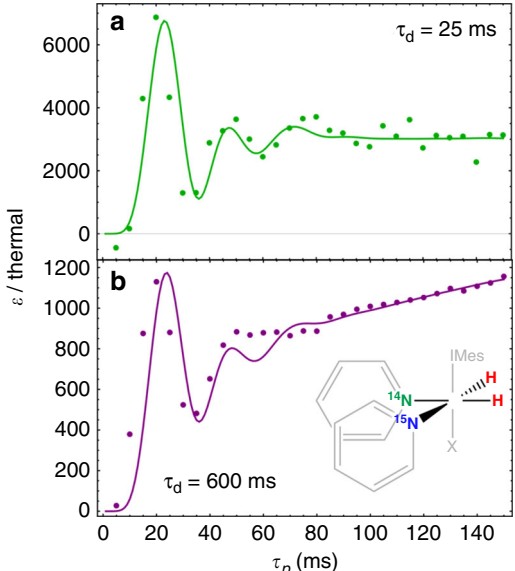

**Fig. 3** Dynamics of isotopically dilute samples. Samples have final pyridine concentrations of 50 mM. **a** DARTH dynamics of natural abundance pyridine with delays $\tau_d = 25$ ms, making the dynamics quasi-CW. **b** DARTH dynamics of 15% labeled pyridine with delays $\tau_d = 600$ ms. Data are shown fit to numerical simulation with an average PTC lifetime of 20 ms

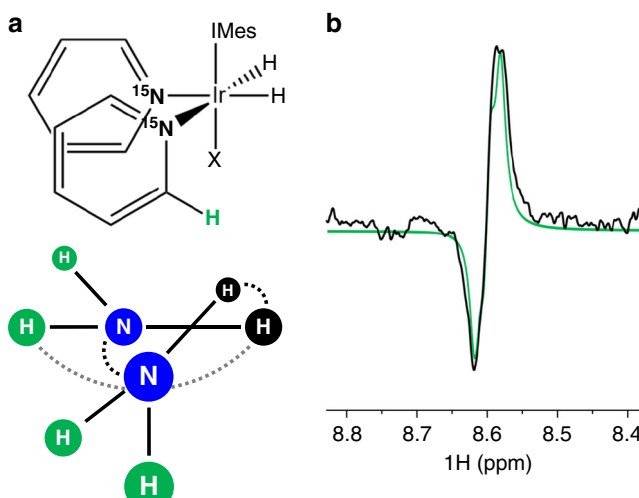

**Fig. 4** Prediction of the ortho-$^1$H resonance lineshape after 25 ms DARTH pulse. **a** Chemical system and magnetic structure used in the simulation along with its dominant homonuclear (dashed, black) and heteronuclear (solid, black) $J$ couplings, as well as the long-range $^4J_{HH}$ coupling (dashed, gray). **b** Experimental specrrum (black) along with fit (green) from QMC simulation

(Fig. 4b). As can be seen from Fig. 4, the theoretical result matches the experimental spectrum with excellent accuracy, emphasizing the robust nature of the QMC simulations.

**SABRE chemical dynamics.** In addition to providing critical insight into the hyperpolarization dynamics of SABRE, these coherent techniques may also yield critical information as to the chemical dynamics of the SABRE processes. This is exemplified here in to ways; firstly, we may utilize a DARTH-pulse to construct a DARTH-EXSY hybrid experiment to directly

measure the hydride dissociation rate (Fig. 4) and we find it to be around 550 ms, more than 10 times the ligand exchange rate. The DARTH pulse generates a hyperpolarized triplet state on the hydrides, which exchanges off the complex during the delay time. By reducing the DARTH-SABRE sequence to a single pulse, one may directly measure the hydride rate of dissociation (Fig. 4b). Measuring the hydride kinetics like this concatenates the inherently second-order kinetics of the hydride exchange and makes it appear as only being first order, as there is distinguishability between the dissociating and associating species.

The pulsed SABRE-SHEATH experiments also provide opportunities to probe the chemical dynamics of the hyperpolarization process. For example, optimization of $\tau_d$ leads to the investigation of aspects of the chemical dynamics that evolve during the delay time. During $\tau_d$, chemical exchange continues and fractionally recharges the singlet state on the hydrides. As apparent from Fig. 5c, $\tau_d$ is experimentally optimized at about 350 ms for experiments where the total length of the pulsing period was kept constant. Note that a sequence of 22 ms on resonance, followed by 2 s off resonance, then repeated many times (a 1% duty cycle) gives more total signal than staying constantly on resonance. However, while keeping the experiment length constant is a reasonable practical comparison, it is much more instructive to look at the enhancement per pulse, which varies by a factor of 47 as the delay is changed (Fig. 5c). For very short times of $\tau_d$, a single complex will experience multiple pulses at the evolution field before dissociation, and since the length was picked for optimal single-pulse excitation the signal is reduced; this effect would be expected to disappear with a higher dissociation rate. Figure 5C also shows that, beyond the ligand exchange rate, there is a second timescale to the delay dynamics, associated with the hydrogen exchange, giving a slowly rising component (at long times) of this data, characterizing the para-H$_2$ regeneration. Accordingly, observing the hyperpolarization dynamics as a function of the storage period $\tau_d$ contributes even more information about the dynamics of the entire system.

## Discussion

We have shown that the DARTH-SABRE and coherent SHEATH experiments boast the ability to monitor the coherent hyperpolarization dynamics under the influence of chemical exchange. Accessing the coherent SABRE dynamics has shown the ability to bypass the damping of the hyperpolarized signal by the SABRE exchange dynamics in certain regimes, which is critical for ligands with exchange rates disparate from the period of their coherent evolution. Moreover, coherent SABRE hyperpolarization has proven to be an ideal model to study quantum systems that evolve under the influence of chemical exchange dynamics, which is readily extensible to many other complex systems, such as the 8-spin AA'(XB$_2$)(X'B'$_2$) system shown here. The implications of the above results are as follows: for a given proposed hyperpolarization substrate, the nuclear spin topology and chemical exchange rate of the ligand will have a considerable effect on the coherent polarization dynamics. The ability to study the coherent polarization dynamics at any field, in turn, allows for a more extensive set of spin topologies and system dynamics to be investigated, each providing a constraint in the design of an optimal hyperpolarization substrate.

## Methods

**Sample preparation and experiment details.** Suitable volumes of a solution of pre-catalyst [Ir(IMes)(COD)Cl] (IMes = 1,3-bis(2,4,6-trimethylphenyl)imidazol-2-ylidene, COD = 1,5-cyclooctadiene) and diluted solutions of pyridine and/or labeled benzonitrile (in methanol-d$_4$) are combined to obtain samples 5 mM in pre-catalyst, 33 mM in pyridine and 100 mM in $^{15}$N-benzonitrile for coherent

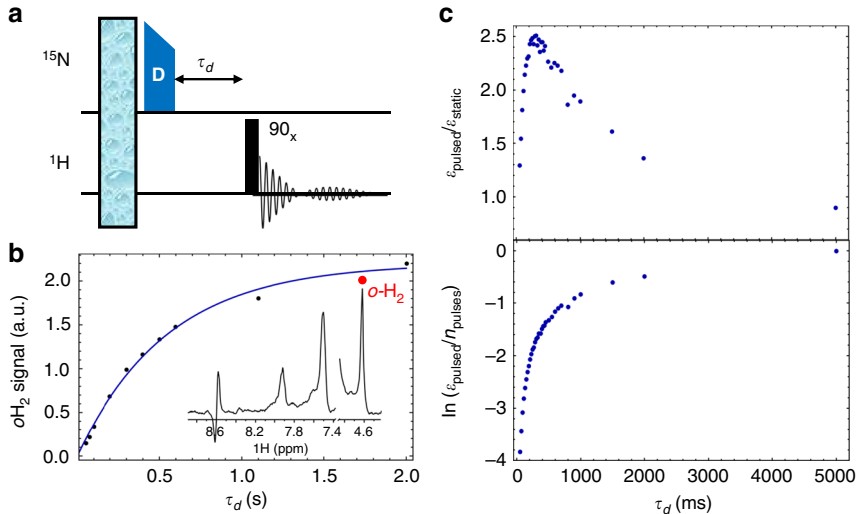

**Fig. 5** Delay dynamics of coherent hyperpolarization experiment. **a** DARTH-EXSY pulse sequence, which uses a pre-saturation bubbling scheme to directly measure exchange of the hyperpolarized hydride. Thermal $T_1$ from inversion recovery is 888.4 ± 2.7 ms. **b** DARTH-EXSY signal on ortho-hydrogen weighted with $T_1$, yielding an average hydride lifetime of 547 ± 32 ms, making singlet order replenishment significantly slower than the ligand exchange. DARTH-EXSY spectrum (single shot, inset) with the signals of hyperpolarized ortho-hydrogen (red) indicated. **c** SHEATH delay dynamics show optimum total signal at $\tau_d = 350$ ms (top plot). Calculating the signal per pulse (bottom plot) also shows the effect of slow hydrogen replenishment with the gradually rising signal at long delay times

SHEATH experiments or samples 4.4 mM in pre-catalyst and 50 mM in pyridine (50 mM, 7 mM, or 0.18 mM in $^{15}$N-pyridine). 500 µL of sample are transferred to a medium wall pressure NMR tube (Wilmad 524-PV-7) and transformed into the catalytically active species with a constant low flow of *para*-hydrogen (50 sccm/min, 45 min at 20 °C and 8 bar). Note that pyridine facilitates activation and acts as a coligand for the PTC complex. Absence of pyridine leads to extremely slow catalyst activation (~72 h at 20 °C) and significantly decreased substrate polarization if benzonitrile is used as a substrate. For all experiments, hyperpolarized signal is detected at 8.45 T.

**Numeric SABRE simulation**. For a given simulation length, a certain number of discrete dissociation events were sampled from a uniform probability distribution. Evolution between these timepoints is dictated only by the Magnus solution to the Liouville von Neumann equation. At the point of dissociation, the dissociation group is selectively traced out of the PTC spin density using tensor contraction in the spin product basis, at which point the singlet order on the parahydrogen is fractionally replenished and a new, unpolarized ligand is introduced to the system. This method maintains coherences between the spins that remain on the PTC. The simulations were averaged over 1600 iterations, which exhibits an error of approximately 1% with respect to a solution exhibiting errors of O(10$^{-7}$), which was taken at 100,000 iterations.

## Data availability
All relevant data shown in the main text and the supplementary information are available from the authors upon request and are used in the example code in the supplementary information. Please contact jacob.lindale@duke.edu or warren.warren@duke.edu for access to the data shown here.

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

## Acknowledgements

The authors thank the following funding sources: NSF CHE-1363008 and CHE-1665090. Research reported in this publication was also supported by the National Institute of Biomedical Imaging and Bioengineering of the NIH under R21EB025313. E.Y.C. additionally thanks the following funding sources: NSF CHE-1416268, NIH 1R21EB020323, and R21CA220137, DOD CDMRP W81XWH-12-1-0159/BC112431 and W81XWH-15-1-0271 awards. J.R.L. and T.T. thank Nathan J. Adamson (Duke University, Department of Chemistry) and Dr. Steven J. Malcolmson for the purification of the [Ir(IMes)(COD)] Cl pre-catalyst.

## Author contributions

J.R.L. designed the DARTH-SABRE experiment, and with S.L.E. collected the experimental data. J.R.L. and W.S.W. conceived and constructed the Quantum Monte Carlo model. C.P.N.T. contributed to the QMC model and, along with Z.Z., J.F.P.C., and G.Z, assisted with experimental data collection. T.T. and E.Y.C. performed initial experiments with ramped pulses. J.R.L., T.T., and W.S.W. wrote the paper.

## Additional information

**Competing interests:** The authors declare no competing interests.

