## [Peer Review File · Nature Communications]

Reviewers' comments:

Reviewer #1 (Remarks to the Author):

The paper is an excellent short communication detailing a considerable breakthrough in the utilisation of parahydrogen spin order for NMR sample polarisation purposes. The work is of sufficient general interest and academic level to merit publication in Nature Communications.

The paper is theoretically and experimentally flawless, I only have a few technical requests:

1. The Hamiltonian in Equations 1 and 2 appears to use angular frequencies as units. The \hbar in Equation 3 should therefore be removed.
2. Dirichlet and Neumann conditions must be better explained and introduced. Even I had to look those up, and it is unclear from the manuscript how they refer to the LvN equation.
3. LvN equation is not elliptic! If only spin degrees of freedom are considered, it is actually an ODE. If the spatial degrees of freedom are also included, then the resulting (Fokker-Planck) equation is parabolic. The entire paragraph is rather obscure and needs to be simplified.
4. The end of the same paragraph is the simple statement to the effect that the ensemble averaging must be applied to the solution rather than the equation of motion. Just say that!
5. Supplementary information is insufficient. Full details of the calculation mathematics and setup process and, ideally, the associated code should be included.

Reviewer #2 (Remarks to the Author):

While the work is undoubtedly of great quality and is a breakthrough in the understanding of the SABRE spin dynamics, I'm sure how accessible this will be to those outside of the SABRE community. In any case, the paper needs substantial revisions, some of which are suggested in my comments.

Specific Comments:

Page 1

Line 24, I believe the acronym is CIDNP and not "CIDNIP"

Page 2

Line 46 "pulsing the excitation field". Which field?

Page 3

Line 50 "and outperform current models ^{^[insert citation(s) here]...}"

Page 7

"would not destroy beta-magnetization" beta-magnetization is not defined.

In Figure 1, to make it easier for the reader, for each part (A, B, C, D) indicate the magnetic fields for the experiment e.g. instead of "A", use "A. High Field" ; "B. Low Field). I had to go back to find this information." Indicate the field ramp to the detection field in the pulse sequence. Particularly in the results in D, the experiment involves the application of the Coherent Sheath sequence is done at a low field followed by the detection at some magnetic field. Maybe I missed it, but it seems the detection field was not specified anywhere in the text or captions, not even in the Experimental Details.

Page 8

Line 94 What do the authors mean by "exchange baseline"? That is not defined anywhere in the text.

Figure C and D. The symbol τ_p on the x-axis of the graphs is never defined. I assumed they are referring to the length of exposure to parahydrogen. Is the concentration of freshly para-enriched H₂ introduced by bubbling assumed to change like a step function? If the concentration is changing due to the finite time the time it takes for the dissolution, wouldn't this have to be convoluted into the kinetic (differential) equations that are solved?

Line 96 “While the enhancements shown here are modest, they become exponentially greater with shorter delay times, producing a maximum enhancement of $\epsilon = 1350$.” The field was not stated so the enhancement factor has no meaning.

Figures C and D, the thermal equilibrium magnetic field for the enhancement factor calculations is never stated.

Line 128

“the quadrupolar relaxation” do they mean the quadrupolar relaxation time?

Overall Impression:

This is really great work! The results presented in Figure 1 are extremely interesting and provided new insights into the dynamics of the SABRE effect in the low and high field regimes. The excellent agreement between experimental data and simulations is excellent. It is very clear from this work that the polarization transfer dynamics can be fully described by coherent dynamics and does not require invoking relaxation superoperator approach as in previous literature. Such insights will undoubtedly be key to the further improvements in hyperpolarization that are attainable, so that the transformation of the singlet order into Zeeman order of the x-nucleus is as close to the ideal efficiency as possible. Enhancement factors close to 5000 were achieved on the ^{15}N nucleus.

I remain confused about the observed enhancement factors. In the abstract. The claim is for the observation of enhancement factor of 1350, yet in Figure 1, I see enhancement factors close to either 500 (for DARTH) and about 4000 for Coherent SHEATH but not 1350.

The authors should explain a bit more about the meaning of the Dirichlet and Neumann conditions and ramifications in the context of the SABRE mechanism and why the relaxation superoperator approach is not valid.

Reviewer #3 (Remarks to the Author):

The manuscript describes how SABRE signal enhancements can be improved by coherent polarization transfer. This has been demonstrated experimentally in high as well as shielded NMR experiments and the match between the developed theory and the experiments is really good. It is shown theoretically and experimentally how the exchange dynamics and the timing of the respective pulse sequences for high or very low magnetic field strength influence each other and how an optimal sequence timing can be used to accomplish higher enhancements. The manuscript substantially improves the current understanding of SABRE NMR experiments and given the importance of this hyperpolarization method is applicable for Nature Communications.

However, there are two major shortcomings of the manuscript which should be ruled out prior to publication:

1. Only polarization transfer to ^{15}N is treated. However, SABRE is a very important method to hyperpolarize also ^{13}C and ^1H nuclei. However, this would lead to much more complex spin systems than the $\text{AA}'\text{XX}'$ and $\text{AA}'\text{X}$ system treated here. The authors should at least speculate if and maybe also how the theory can be extended to cover more complex spin systems and how ^{13}C and ^1H hyperpolarization by SABRE can be improved using their approach.
2. It was shown that the main drawback of SABRE, namely its limitation to specific substrates, can be improved by adding coligands to the SABRE mixture (J. Am. Chem. Soc., 2014, 136 (7), pp 2695–2698). The authors should comment on if and maybe also how the effect of coligand binding can be included in their theory.

Minor issues:

- The title suggests that the manuscript deals with every hyperpolarization technique, please modify it by adding "SABRE" somewhere, e.g. Unveiling Coherently-Driven SABRE Hyperpolarization Dynamics
- Reference 20 and 43 are the same
- Page 2, 5th line from the bottom: I do not understand what is meant by "linewidth from dissociation". Please explain that more explicitly.
- Page 4, beginning of "Coherent Hyperpolarization Dynamics" section: it would be easier for the reader to follow if you introduce the investigated spin systems in the beginning of this section: IMes-catalyst with ^{15}N enriched pyridine for DARTH-SABRE and IMes-catalyst with ^{15}N enriched benzonitrile for coherent SHEATH. Why were both experiments not performed on both substrates? That would significantly help to verify the general validity of your approach.
- Page 8, end of first paragraph: if much higher enhancements are achievable by shorter delay times why this was not experimentally demonstrated? This would have been much more convincing than showing the moderate enhancements in Figure 1.

- Page 8, last paragraph: I cannot find a dashed line in figure 1E. (I was reading the document named: 180236_0_art_file_3244726_pdzbns.docx)
- Caption of Figure 3: the ortho-proton on pyridine is labeled with a purple dot not with a green one, as the caption says
- Page 14, second line: should be “total length”
- Page 14, third sentence of the conclusion “Moreover, coherent SABRE hyperpolarization has proven to be an ideal model to study quantum systems that evolve under the influence of chemical exchange dynamics, which is readily extensible to many other complex systems.” This would be more convincing if the authors can provide some examples for the “many other complex systems”.

Answers to Remarks: Reviewer #1

The paper is an excellent short communication detailing a considerable breakthrough in the utilisation of parahydrogen spin order for NMR sample polarisation purposes. The work is of sufficient general interest and academic level to merit publication in Nature Communications.

The paper is theoretically and experimentally flawless, I only have a few technical requests:

Thank you very much for your favorable review! I have answered the 5 technical points that you mention in your review below.

1. The Hamiltonian in Equations 1 and 2 appears to use angular frequencies as units. The \hbar in Equation 3 should therefore be removed.

Factor of \hbar removed from equation 3.

2. Dirichlet and Neumann conditions must be better explained and introduced. Even I had to look those up, and it is unclear from the manuscript how they refer to the LvN equation.

3. LvN equation is not elliptic! If only spin degrees of freedom are considered, it is actually an ODE. If the spatial degrees of freedom are also included, then the resulting (Fokker-Planck) equation is parabolic. The entire paragraph is rather obscure and needs to be simplified.

4. The end of the same paragraph is the simple statement to the effect that the ensemble averaging must be applied to the solution rather than the equation of motion. Just say that!

2.-4. The paragraph in question has been largely re-written for clarity and conciseness. Thank you for the input on the error regarding the LvN equation.

5. Supplementary information is insufficient. Full details of the calculation mathematics and setup process and, ideally, the associated code should be included.

5. The code has been implemented in the *RogueSpin* package in Mathematica, which was built by J.R.L. to perform these QMC simulations. SI now contains full example of QMC code along with the mathematics and setup required to perform the fits shown in the article. Furthermore, the entire QMC engine is included along with a step-by-step introduction to running QMC simulations.

Answers to Remarks: Reviewer #2

While the work is undoubtedly of great quality and is a breakthrough in the understanding of the SABRE spin dynamics, I'm sure how accessible this will be to those outside of the SABRE community. In any case, the paper needs substantial revisions, some of which are suggested in my comments.

Thank you for your detailed review! We have answered your remarks and questions below:

Page 1, Line 24, I believe the acronym is CIDNP and not "CIDNIP"

1. *Page 1 Line 24*: Thank you for the correction!

Page 2 Line 46 "pulsing the excitation field". Which field?

2. *Page 2 Line 46*: By "excitation field", we mean the field generating the hyperpolarized signal. This has been changed in the text accordingly.

Page 3, Line 50 "and outperform current models ^{^[insert citation(s) here]...}"

3. *Page 3 Line 50*: Citations moved.

Page 7 “would not destroy beta-magnetization” beta-magnetization is not defined.

4. *Page 7 “beta-magnetization not defined”*: this sentence has been re-structured for clarity.

In Figure 1, to make it easier for the reader, for each part (A, B, C, D) indicate the magnetic fields for the experiment e.g. instead of “A”, use “A. High Field” ; “B. Low Field). I had to go back to find this information.” Indicate the field ramp to the detection field in the pulse sequence. Particularly in the results in D, the experiment involves the application of the Coherent Sheath sequence is done at a low field followed by the detection at some magnetic field.

5. *Page 7 Figure 1 comments*: All of the Figure 1 comments (now Figure 2) have been addressed.

Maybe I missed it, but it seems the detection field was not specified anywhere in the text or captions, not even in the Experimental Details

6. *Detection Field*: All of the hyperpolarized signals are detected at 8.45 T, and thus the enhancements are referenced to thermal signal at this field. This has been clarified in the figure and also the experimental details.

Page 8 Line 94 What do the authors mean by “exchange baseline”? That is not defined anywhere in the text.

7. *Page 8 Line 94*: “Exchange baseline” is the polarization to which the dynamics converge under steady-state conditions. This has been defined in the text.

Figure C and D. The symbol τ_p on the x-axis of the graphs is never defined. I assumed they are referring to the length of exposure to parahydrogen. Is the concentration of freshly para-enriched H₂ introduced by bubbling assumed to change like a step function? If the concentration is changing due to the finite time it takes for the dissolution, wouldn't this have to be convoluted into the kinetic (differential) equations that are solved?

8. τ_p : This parameter is the length of the coherent DARTH or SHEATH pulse, not the length of exposure to para-hydrogen. The sample is bubbled continuously throughout the experiment, but the dynamics are detected by scanning this parameter.

Line 96 "While the enhancements shown here are modest, they become exponentially greater with shorter delay times, producing a maximum enhancement of $\epsilon = 1350$." The field was not stated so the enhancement factor has no meaning. Figures C and D, the thermal equilibrium magnetic field for the enhancement factor calculations is never stated.

9. *Page 8 Line 96*: The value of a 1350x enhancement was reported in the text with the full data set given in the SI, but we have moved the data set to Figure 1 (now Figure 2E) after the revisions. Again, the enhancements are all given at 8.45 T.

Line 128 "the quadrupolar relaxation" do they mean the quadrupolar relaxation time?

10. *Line 128*: We do intend for this to be the "quadrupolar relaxation time" and it is changed in the text.

Overall Impression:

This is really great work! The results presented in Figure 1 are extremely interesting and provided new insights into the dynamics of the SABRE effect in the low and high field regimes. The excellent agreement between experimental data and simulations is excellent. It is very clear from this work that the polarization transfer dynamics can be fully described by coherent dynamics and does not require invoking relaxation superoperator approach as in previous literature. Such insights will undoubtedly be key to the further improvements in hyperpolarization that are attainable, so that the transformation of the singlet order into Zeeman order of the x-nucleus is as close to the ideal efficiency as possible. Enhancement factors close to 5000 were achieved on the ^{15}N nucleus.

I remain confused about the observed enhancement factors. In the abstract. The claim is for the observation of enhancement factor of 1350, yet in Figure 1, I see enhancement factors close to either 500 (for DARTH) and about 4000 for Coherent SHEATH but not 1350.

The authors should explain a bit more about the meaning of the Dirichlet and Neumann conditions and ramifications in the context of the SABRE mechanism and why the relaxation superoperator approach is not valid.

11. *Dirichlet/Neumann & why relaxation superoperator approach is not valid:* This section has largely been re-written for clarity and to emphasize why discretized methods are required under this intermediate limit when $J_{\text{NH}} \sim k_d$.

Answers to Remarks: Reviewer #3

The manuscript describes how SABRE signal enhancements can be improved by coherent polarization transfer. This has been demonstrated experimentally in high as well as shielded NMR experiments and the match between the developed theory and the experiments is really good. It is shown theoretically and experimentally how the exchange dynamics and the timing of the respective pulse sequences for high or very low magnetic field strength influence each other and how an optimal sequence timing can be used to accomplish higher enhancements. The manuscript substantially improves the current understanding of SABRE NMR experiments and given the importance of this hyperpolarization method is applicable for Nature Communications.

However, there are two major shortcomings of the manuscript which should be ruled out prior to publication:

Thank you very much for your favorable review! Your comments for major revisions were actually very simple to address and add a complementary dimension to the article. We will begin by addressing these, and then will address the comments for minor revisions.

Only polarization transfer to ^{15}N is treated. However, SABRE is a very important method to hyperpolarize also ^{13}C and ^1H nuclei. However, this would lead to much more complex spin systems than the $\text{AA}'\text{XX}'$ and $\text{AA}'\text{X}$ system treated here. The authors should at least speculate if and maybe also how the theory can be extended to cover more complex spin systems and how ^{13}C and ^1H hyperpolarization by SABRE can be improved using their approach.

1. *Polarization transfer to $^{13}\text{C}/^1\text{H}$ and extensibility of the simulation method to more complex spin systems other than the $\text{AA}'\text{XX}'$ and $\text{AA}'\text{X}$ systems.*

This was an excellent comment, as it let us really emphasize the diversity of systems that may be studied with this method and led to an additional paragraph in the manuscript. As the simulation utilizes only coherent dynamics to perform propagation during the lifetime of each PTC, the construction of more complicated systems truly only requires 1) constructing a more complicated spin Hamiltonian for the larger system (at the moment limited to 15 spins due to dimensionality restriction of exponentiating non-sparse matrices in Hilbert space) and 2) ensuring that all of the spins belonging to a single ligand 'dissociate' simultaneously. As such, it is trivial to study the dynamics of any arbitrary nucleus (and, in fact, the simulation package can handle any spin-1/2, spin-1, or spin 3/2 nucleus).

To show this, we use the QMC simulations to predict the lineshape of the ortho- ^1H on the pyridine ring after a 25 ms DARTH pulse, with no *a priori* state information given to the simulation other than the initial state of the para-hydrogen and unity on all other spins. While the $[\text{Ir}(\text{H})_2(\text{IMes})(^{15}\text{N-pyr})_3]^+$ complex is reduced to an $\text{AA}'\text{XX}'$ spin system by the symmetry of the ortho- ^1H on the pyridine ring, one must expand the spin system to an 8-spin $\text{AA}'(\text{XB}_2)(\text{X}'\text{B}'_2)$ system to describe the dynamics of the ^1H . We show excellent agreement between the QMC solution and the experiment (new Figure 4 copied below for reference).

Figure 4 | Prediction of the ortho- ^1H resonance line-shape after 25 ms DARTH pulse. **A** Chemical system and magnetic structure used in the simulation along with its dominant homonuclear (dashed, black) and heteronuclear (solid, black) J -couplings as well as the long-range $^4J_{\text{HH}}$ coupling (dashed, grey). **B** Experimental spectrum (black) along with fit (green) from QMC simulation.

It was shown that the main drawback of SABRE, namely its limitation to specific substrates, can be improved by adding coligands to the SABRE mixture (J. Am. Chem. Soc., 2014, 136 (7), pp 2695–2698). The authors should comment on if and maybe also how the effect of coligand binding can be included in their theory.

2. *Co-ligand effects in the simulation.*

This was another excellent comment, and in fact, the data in the paper already allowed us to discuss both types of effects generated by co-ligands: ligands that affect only the chemical dynamics of the PTC and ligands that affect the coherent dynamics of the PTC as well. The response to this remark was built into the **Coherent Hyperpolarization Dynamics** section of the paper at various points.

In the first case, the AA'X SHEATH system is an example of one where the pyridine co-ligand does not contribute to the coherent dynamics of the system, and only alters the exchange rate of the benzonitrile, as we do not require the inclusion of the co-ligand to achieve excellent agreement with experiment and the inclusion would alter the coherent dynamics. These types of co-ligands will only have the effect of changing the exchange rate, which can be simulated by *ab initio*/DFT electronic structure calculations.

In the second case, the AA'XQ system highlights how the simulation is handled when the co-ligand (in this case an isotopic “co-ligand”) affects the coherent dynamics. In this case, the spins of the co-ligand must be simply included in the calculation, and given a known exchange rate (which in this case is the same as the exchange rate from the fully labelled version), one can accurately model the dynamics with an asymmetric co-ligand.

Naturally, there is a third case, which combines the first two cases (alters the exchange rate *and* the coherent dynamics), but this is a trivial extension of the second case, simply with a different exchange rate. Again, this is trivial only because evolution is carried out fully coherently.

Minor issues:

The title suggests that the manuscript deals with every hyperpolarization technique, please modify it by adding “SABRE” somewhere, e.g. Unveiling Coherently-Driven SABRE Hyperpolarization Dynamics

1. *Title:* The title has been changed to “Unveiling Coherently-Driven SABRE Hyperpolarization Dynamics”.

Reference 20 and 43 are the same

2. *Reference:* Reference 43 has been omitted.

Page 2, 5th line from the bottom: I do not understand what is meant by “linewidth from dissociation”. Please explain that more explicitly.

3. *Page 2 Line 5 from Bottom:* “Linewidth from dissociation” is the width of the exchange-broadened resonance of the active complex. This has been changed in the manuscript for clarity.

Page 4, beginning of “Coherent Hyperpolarization Dynamics” section: it would be easier for the reader to follow if you introduce the investigated spin systems in the beginning of this section: IMes-catalyst with ^{15}N enriched pyridine for DARTH-SABRE and IMes-catalyst with ^{15}N enriched benzonitrile for coherent SHEATH. Why were both experiments not performed on both substrates? That would significantly help to verify the general validity of your approach.

4. *Introduce the investigated spin systems & why we do not show both experiments with both complexes:* The chemical systems and their magnetic structure are introduced in an initial figure (now Figure 1). Both experiments were not performed on both complexes as SLIC-based hyperpolarization techniques (LIGHT-SABRE, RF-SABRE, DARTH-SABRE, etc.) require chemically equivalent para-hydrogen derived hydrides. However, there is ~ 2 ppm separation between the hydrides in the $[\text{Ir}(\text{H})_2(\text{IMes})(^{15}\text{N}\text{-PhCN})(\text{pyr})_2]^+$ complex, making it impossible to study 3-spin systems under a DARTH pulse unless they are 3-spin systems by isotopically labelling (forcing chemical equivalence of the hydrides), which is why we show the natural abundance and fractionally enriched pyridine samples under various conditions that verify the validity of the simulations. We have collected data on the $[\text{Ir}(\text{H})_2(\text{IMes})(^{15}\text{N}\text{-pyr})_3]^+$ SABRE system under coherent SHEATH conditions, but the coupling to the ^1H pyridine protons causes fast T_1 relaxation on the ^{15}N , making it very difficult to collect clean data without an automated shuttling system. The ^{15}N -benzonitrile T_1 relaxation is significantly longer and is not coupled to any protons, making it very easy to collect clean data.

Page 8, end of first paragraph: if much higher enhancements are achievable by shorter delay times why this was not experimentally demonstrated? This would have been much more convincing than showing the moderate enhancements in Figure 1.

5. *Page 8 end of first paragraph:* We have shown the enhancement as a function of the delay length given a 25 ms DARTH pulse in (now) Figure 2. We show the moderately enhanced data in Figure 2 because this is the simplest DARTH condition, as a proof of validity, but follow this up by exploring the effect of delay length on the coherent dynamics in the natural abundance section. Specifically, we use the optimal delay length (25 ms) to generate the quasi-CW condition in the dynamics, which gives a ~7000x enhancement for natural abundance pyridine at the maximum and is clearly a factor of 2 greater than the exchange baseline/steady state polarization.

Page 8, last paragraph: I cannot find a dashed line in figure 1E. (I was reading the document named: 180236_0_art_file_3244726_pdzbn.docx)

6. *Page 8 last paragraph:* This is an error from a previous revision, apologies.

Caption of Figure 3: the ortho-proton on pyridine is labeled with a purple dot not with a green one, as the caption says

7. *Fixed error in caption*

Page 14, second line: should be "total length"

8. *Page 14 second line:* Changed typo.

Page 14, third sentence of the conclusion “Moreover, coherent SABRE hyperpolarization has proven to be an ideal model to study quantum systems that evolve under the influence of chemical exchange dynamics, which is readily extensible to many other complex systems.” This would be more convincing if the authors can provide some examples for the “many other complex systems”.

9. *Third sentence of conclusion:* I now reference the paragraph generated by your major revision point #1 to emphasize the range of complex systems that one can simulate.

REVIEWERS' COMMENTS:

Reviewer #1 (Remarks to the Author):

All of my comments have been addressed and the paper is good to fly. The authors should add the missing pages from the Mathematica printout in the supplementary information and fix the SI title.

Reviewer #2 (Remarks to the Author):

The major criticisms as well as the minor corrections have been addressed, and the quality of the paper has been significantly improved. I have no further comments. The paper is suitable for publication in Nature Comm.

Reviewer #3 (Remarks to the Author):

I am really happy with the revised version of the manuscript and have no further comments. The authors explicitly addressed each point raised by the reviewers and the manuscript is now ready for publication.